# The Effects of Roadside Woody Vegetation on the Surface Temperature of Cycle Paths

Nikola Žižlavská, Tomáš Mikita * and Zdeněk Patočka

Forest Management and Applied Geoinformatics, Faculty of Forestry and Wood Technology, Mendel University in Brno, 613 00 Brno, Czech Republic; nikola.zizlavska@mendelu.cz (N.Ž.); zdenek.patocka@mendelu.cz (Z.P.)
* Correspondence: tomas.mikita@mendelu.cz

**Abstract:** The article is on the effects of woody vegetation growing on the roadside on the temperature of the surface of cycle paths. The main hypothesis of the study is that vegetation has the effect of lowering the temperature of the surroundings in its shadow and thus improves the comfort of users of cycle paths in the summer months. The second hypothesis is to find out which type of road surface is most suitable for the thermal well-being of users. This goal was achieved by measuring the temperature of selected locations on cycle paths with different types of construction surfaces with nearby woody vegetation using a contactless thermometer over several days at regular intervals. The positions of the selected locations were measured using GNSS and the whole locality of interest was photographed using an unmanned aerial vehicle (UAV), or drone, and subsequently a digital surface model (DSM) of the area was created using a Structure from Motion (SfM) algorithm. This model served for the calculation of incident solar radiation during the selected days using the Solar Area Graphics tool with ArcGIS software. Subsequently, the effect of the shade of the surrounding vegetation on the temperature during the day was analysed and statistically evaluated. The results are presented in many graphs and their interpretation used to evaluate the effects of nearby woody vegetation and the type of road surface on the surrounding air temperature and the comfort of users of these routes. The results demonstrate the benefits of using UAVs for the purpose of modelling the course of solar radiation during the day, showing the effect of roadside vegetation on reducing the surface temperature of the earth's surface and thus confirming the need for planting and maintaining such vegetation.

**Keywords:** surface temperature; air temperature; woody roadside vegetation; road surface; digital surface model; Structure from Motion; UAV; cycle routes; Czech Republic





## 1. Introduction

Urban areas are expanding faster and faster in the landscape [1]. The consequence of this growth is increasing pressure to build green infrastructure, not only by providing basic ecosystem services, but also to improve human thermal comfort [2]. Urban areas maintain a specific climate, which is characterised by a significantly warmer environment than the surrounding landscape [3] and differs significantly in meteorological conditions from the countryside. In the case of air temperature, the result is the urban heat island (UHI) [3,4], often considered to be a major factor influencing suburban and urban climates [5,6]. During the day, heat is received as solar radiation, which is absorbed by the ground surface regardless of its type (without vegetation, with vegetation, water surface and so on) [7]. Part of the solar radiation is reflected by the ground surface and part is absorbed into it, so it is warmed [8]. During the day, the minimum temperature occurs immediately after sunrise and, conversely, at 1 pm the maximum occurs, because after this time the heat output exceeds its intake. During the day, the ground surface is warmer than the overlying layers of air, and at night it is the other way around. The biggest difference is the temperature between the ground surface and air in summer. A completely different

situation occurs at night when heat is no longer received from the sun while the loss of heat by radiation continues. At this point, the ground surface is colder than the space above it [7].

The effect of urban heat islands in cities is not, in principle, a new phenomenon. The phenomenon was recognised in the last century as a global problem [9] and it was found that global warming was progressing faster than predicted [10]. In large cities, the temperature difference is around 10 °C; in smaller cities it is around 5 °C, but with growing urban areas, it is rapidly gaining importance. The metropolis creates artificial islands, which are formed by buildings, roads, pavements and the large roofs of shopping malls and industrial buildings [11].

Urban ecosystems can be characterised by two types of surface: paved and vegetated surfaces [12]. Paved surfaces increase the temperature and endanger trees with drought due to the absorption and storage of solar energy during the day, while vegetation cover has a cooling effect during the day due to evapotranspiration and shading [13]. Paved areas in an area, such as buildings, roads etc., have a similar effect.

Materials such as concrete and asphalt can absorb much more heat from the sun than natural materials found in rural areas. These materials dominate in cities and thus contribute to the absorption and storage of solar energy during the day. During a night without sunlight, heat should be radiated from surfaces back into the atmosphere as long-wave radiation and thus significantly affects the air temperature [7]. In balance with paved surfaces, vegetation cover—thus green infrastructure—should be designed to reduce the effects of heat islands [14]. Dark materials, together with the placement of urban areas and roads, which are close together and almost free of vegetation, also contribute to heat absorption during the day, when solar radiation is the main factor influencing air temperature. Surface temperatures differ according to the materials used, such as asphalt and concrete, and this is the main factor influencing the ambient temperature of these surfaces and their overall temperature balance [15].

Due to its dark colour, asphalt has a greater ability to absorb sunlight than to reflect it, thus increasing both its temperature and that of the surrounding air. In contrast, a concrete surface reflect rather than absorb sunlight due to a lighter colour. In general, these surfaces absorb and store more thermal energy during the day, which they then release back into the atmosphere during the night [7].

Conversely, vegetated surfaces can help mitigate high urban temperatures by providing shade from the sun and cooling by evapotranspiration. In larger urban areas, anthropogenic heat increases evaporation and thus cooling decreases natural ventilation changes and thus the overall heat balance in cities changes [16].

Paved surfaces in the countryside, such as roads across fields and forests, and cycle paths, have a similar effect on the ambient temperature, as they warm the surrounding environment and, above all, affect the users of these roads. Trees are often planted along country roads because, in addition to their aesthetic value, they reduce the surface temperature.

Woody roadside vegetation—green infrastructure—performs a specific and irreplaceable function in the cycle of substances in the landscape. It accelerates rock weathering, contributes to soil formation and development, prevents erosion [17], mitigates temperature extremes [18], regulates evaporation and the overall water cycle of the landscape [7], captures particles of dust, absorbs carbon dioxide and releases oxygen [17]. Vegetation also has a great influence on the surrounding microclimate. Temperature reduction by vegetation cannot be explained by individual factors; it is a combination of various functions such as evapotranspiration [19], maintaining moisture and covering the surface [20]. It can also reduce noise [21], reduce mental stress [22,23], improve air quality [24], enhance the thermal comfort of people [25] and increase the involvement of people in cities due to the friendly environment it creates [26,27]. Seven environmental factors affect human energy: air temperature, air movement, humidity, sunlight, surface radiation, metabolic heat and clothing insulation [28,29]. The temperature in the shade of a tree can be up to 3 °C lower on a sunny summer day than the temperature in the shade of an inanimate object, such as

an umbrella. Green infrastructure evaporates water, which consumes heat energy from the surroundings [30]. Thus, vegetation not only compensates for temperature fluctuations but also has a positive effect on air humidity, thus contributing to favourable air quality for human health and improving the thermal comfort of people [31].

A surface covered with vegetation does not heat up as a paved or bare surface does, because in green plants, algae, cyanobacteria and some bacteria a complex biochemical process called photosynthesis takes place. More specifically, green plants use light energy in the visible part of the solar spectrum (wavelengths 400 to 700 nm) to convert simple molecules into complex organic substances such as starch, proteins and oils. It is by capturing light with a pigment that photosynthesis, which consumes the energy of light radiation, can takes place. Because light energy is intercepted by plants, the surface temperature beneath them is decreased [32].

Methods for measuring surface temperature include thermal imaging, either using hand-held cameras or remote sensing (RS) technologies such as satellite, aerial imaging or Unmanned Aerial Vehicle (UAV) (so called drone) imaging. Thermal electromagnetic radiation includes wavelengths from 3 μm to 12 μm (atmospheric windows in the intervals of 3–5 μm and 8–12 μm). At these wavelengths, the intensity of electromagnetic radiation is very low and these wavelengths are not visible to the human eye [33].

Among the most widely used satellite systems for thermal imaging are the Landsat 7 ETM+ and Landsat 8 OLI/TIRS, Terra/Aqua Moderate Resolution Imaging Spectroradiometer (MODIS) and Advanced Spaceborne Thermal Emission and Reflection Radiometer (ASTER) sensors [34]. Landsat 7 ETM+ covers the thermal band with 60 m spatial resolution and captures data between 10.40 and 12.50 μm [35]. Landsat 8 OLI/TIRS covers two thermal bands with 100 m spatial resolution in 10.60–11.19 μm and 11.5–12.51 μm [36]. MODIS has six bands in one atmospheric window from 3.66 to 4.55 μm and 10 bands from 6.535 to 14.385 μm [37]. The spatial resolution of all these bands is 1 km. ASTER has five bands from 8.125 to 11.65 μm with 90 m spatial resolution [38]. In addition to low spatial resolution, low temporal resolution is also a disadvantage (16 days of the grounding track repeat cycle in the cases of both Landsat and ASTER and 1–2 days for Terra/Aqua MODIS). The low spatial and temporal resolution of the thermal images is not applicable for the purpose of evaluating the influence of vegetation on the surface temperature at the local level.

Drones are commonly equipped with GNSS and digital cameras. Drones combine high spatial resolution with easy operability and low operational cost [39]. They can be rotor-based, fixed-wing or hybrid [40] and are generally equipped with high-resolution digital cameras, multispectral cameras, lidar (Light Detection and Ranging) or infrared/thermal cameras. Thermal cameras operate in the spectrum from 8000 to 15,000 nm and their temperature reading can be derived from the pixel number [41]. Miniaturised thermal infrared (TIR) cameras can be divided into those that are radiometrically and non-radiometrically calibrated [42]. Non-radiometrically calibrated cameras can only provide information on relative temperature differences which are expressed in raw digital numbers (DN, representing the magnitude of the TIR radiance). However, measurements with radiometric cameras are subject to error and some factors must be considered. In addition to internal factors which are solved by non-uniformity corrections (NUC) [43], there are external factors such as target object emissivity, ambient humidity, the temperatures of surrounding objects or the sky, and the distance between the camera and the target [44]. Although it is possible to obtain detailed thermal images of the surface by near-drone imaging, drone data are not suitable for temperature monitoring, as it requires repeated imaging during the day, which is time consuming. In addition, the terrain under the treetops is not visible when shooting with drones.

Personal thermal comfort is a state of mind that expresses satisfaction with the environs. High temperatures and humidity cause feelings of discomfort that can lead directly to thermal stress. However, it is important to mention that people react differently to these influences depending on their physical and mental health [45].

Although temperature reduction due to vegetation shading is well known, it is very difficult to quantify this phenomenon and find out how vegetation along roads helps to reduce the surface temperature and the thermal comfort of users. Therefore, the goal was to determine the influence of different types of forest and field road surfaces on the thermal comfort of users (pedestrian, cyclist) and, at the same time, to determine the influence of the shade of roadside vegetation on reducing surface temperature. Both goals were achieved by temperature measurement of different surfaces with subsequent analysis of sun course and shadow cast by vegetation.

## 2. Materials and Methods

The site is located in the south-eastern part of the Czech Republic, in the cadastral area of the town Jedovnice, which lies about 15 km northeast of the city of Brno (Figure 1).

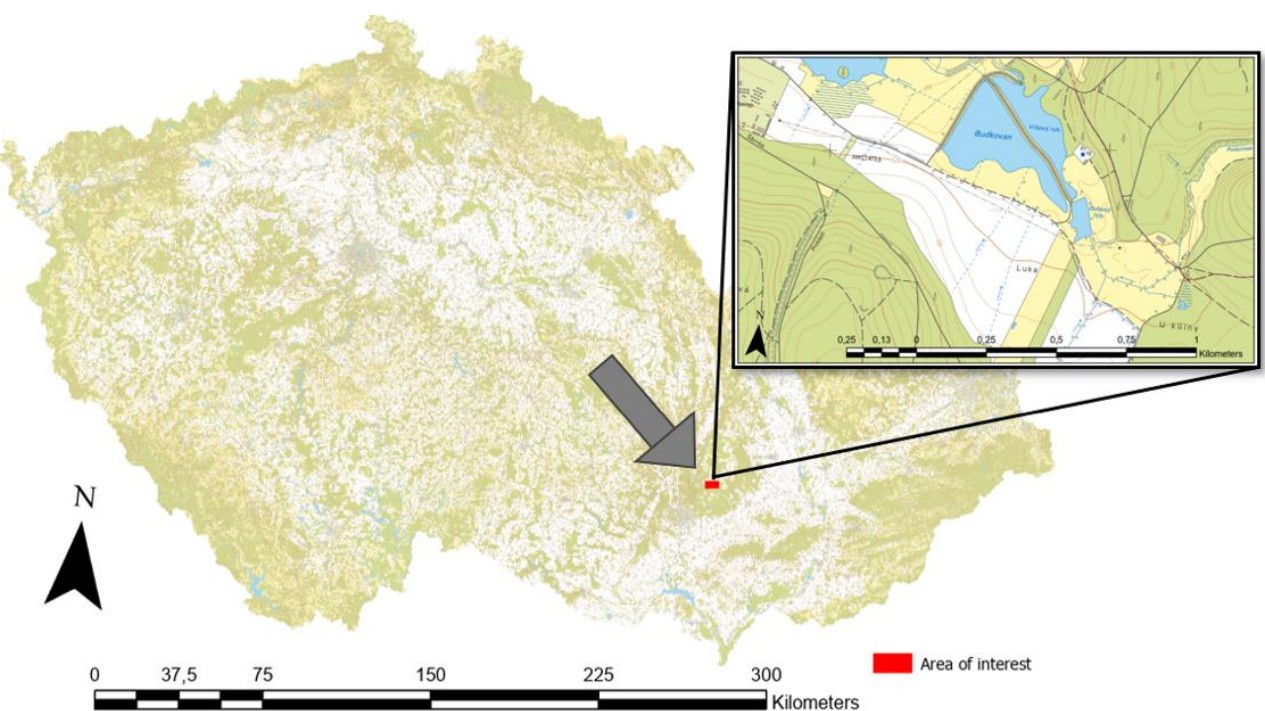

**Figure 1.** Area of interest within the Czech Republic, source [46].

The area is in the 4th vegetation stage, which is the most widespread in the Czech Republic, covering 42.6% of the area [47,48]. Average temperatures here from June to September are around 19.4 °C [49]. In the warm half of the year, the period from April to September, approximately 327 mm of precipitation falls here, around 61% of the annual amount, and the maximum ten-year average monthly precipitation totals are in June to August [49].

A total of 18 localities were selected for temperature measurement and continuous measurements were taken at hourly intervals (Figure 2). The measured locations had different types of surface and were in sunny or shaded places (Table 1). An important factor was to find different surfaces for individual groups of measured locations relatively close to each other because of the time required for measurement (measurement was planned after one hour) and so that at different places for groups of measured locations there were different lighting conditions (usually two points—light/shadow).

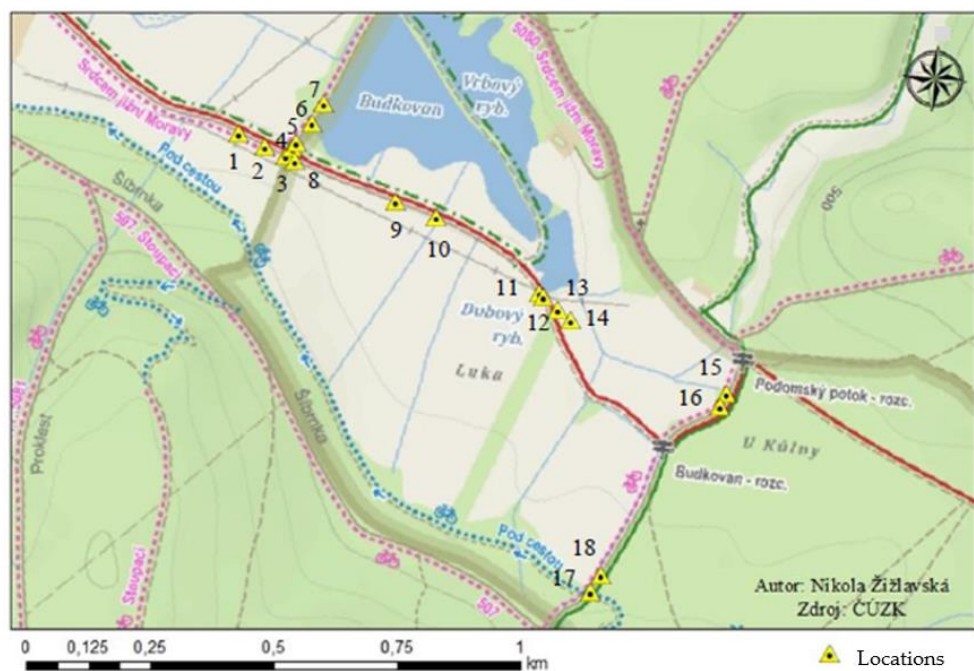

**Figure 2.** Distribution of measured locations in the area of interest, source [46,50].

**Table 1.** Types of measured surfaces.

| Location | Surface | Aspect |
|:---:|:---|:---:|
| 1 | asphalt mixture | SE |
| 2 | asphalt mixture | SW |
| 3 | asphalt mixture | SE |
| 4 | gravel abrasive layer of crushed aggregate fraction 8/16 mm on the subgrade, which is operationally strengthened | SE |
| 5 | gravel abrasive layer of crushed aggregate fraction 8/16 mm on the subgrade, which is operationally strengthened | SE |
| 6 | gravel abrasive layer of crushed aggregate fraction 8/16 mm on the subgrade, which is operationally strengthened | SE |
| 7 | gravel abrasive layer of crushed aggregate fraction 8/16 mm on the subgrade, which is operationally strengthened | SE |
| 8 | crumbling abrasive layer of asphalt mixture | NW |
| 9 | operational reinforcement | S |
| 10 | operational reinforcement | NW |
| 11 | ballast made of quarry stone | E |
| 12 | ballast made of quarry stone | N |
| 13 | natural ground | NW |
| 14 | natural ground | S |
| 15 | vibrated gravel on wood chips | E |
| 16 | vibrated gravel on wood chips | SE |
| 17 | vibrated gravel | NW |
| 18 | vibrated gravel | SW |

### 2.1. Data Collection

Surface temperatures were measured on 8 August, 18 August, 20 August, 4 September and 12 September 2019 using a Bosch PTD 1 hand-held non-contact thermometer. This thermometer does not allow direct adjustment for the emissivity of the measured surface; however, it is possible to choose temperature measurement with a low, medium or high emissivity. All types of selected surface had high emissivity in the range 0.9 to 0.95 and therefore the high emissivity setting was chosen for all measurements at all locations. The surface temperature was measured at intervals of one hour and proceeded from point

No. 1 to point No. 18. The surface temperature, air temperature and vegetation surface temperature were measured. The vegetation surface temperature was measured 1 m from the edge of the road on permanent grassland. The measurements always started on the hour at point No. 1 and the last point No. 18 was measured about 30 min later. This time delay was due to the area over which the measured locations were distributed.

During these five days, measurements were made at all 18 locations as described above from 10:00 or 11:00 h to 16:00 or 17:00 h. Then, on 5 September 2019, measurements were performed only at locations No. 1 and No. 3, where the air temperature and the surface temperature of the asphalt mixture were measured from 8:30 to 20:00 h. On 12 September 2019, at 14:00 h, in addition to non-contact surface temperature measurements, the area was also photographed with an RGB camera and a thermal camera from a height of 80 m using a drone to create a digital model of the surface and temperature images.

## 2.2. RGB and Thermal Imaging

To create a thermal map of the area, and in particular a digital model of the surface, an unmanned device DJI Mavic 2 Enterprise Dual, equipped with an integrated non-radiometric (imaging) thermal camera FLIR Lepton 3.5 with a resolution of 0.3 Mpix and an RGB camera with a resolution of 12 Mpix, was used. The non-radiometric camera of this drone stores only thermal images in the form of a three-band image. The output is an image with a defined colour palette.

The parameters of the radiometric thermal camera Flir Lepton 3.5 on drone DJI Mavic 2 Enterprise Dual are:

- Image resolution: $640 \times 480$ pixels
- Wavelength: 8–14 μm
- Minimum resolution: 0.05 °C

The imaging itself was performed based on a predefined flight plan with an overlap in the rows of 90% and an overlap of slides in the row of 90%. In one flight, 270 images were taken from a height of 80 m above the ground.

## 2.3. Data Processing and Evaluation

Data processing was performed using Microsoft Excel, AGISOFT Metashape Professional, Esri ArcGIS and STATISTICA 12 software. Based on the measured data, graphic outputs (maps, pictures, digital terrain model, etc.), tables and graphs were made. The images from the drone were processed using AGISOFT Metashape Professional software into a point cloud and then a digital model of the surface and an orthophoto with a resolution of 0.05 m in the visible spectrum and a thermal image with a resolution of 0.25 m. The thermal map was created based on the calibration of the colour scale using the temperatures measured at individual points in the software ArcGIS 10.7.1.

One of the goals was to find the influence on the surface temperature of the woody vegetation near the cycle routes. Thanks to the created DSM (Figure 3), it was possible to determine the course of solar radiation during the days when the temperature was measured, and thus determine the dependence of the surface temperature on sun exposure or shading of the monitored localities.

For this purpose, the Solar Radiation Graphics tool of the ArcGIS software was used, which, based on DSM, allows the shading of the horizon above a given point to be calculated as well as the course of the sun during the selected day (Figure 4). By combining the shading of the horizon and the course of the sun, it is possible not only to find out at what time the given point was shaded or sunlit but also to calculate the total time during which the point was in the shade or the sun. However, the tool counts on DSM as impermeable, so it does not take into account, for example, partial shade and partial incident radiation through the treetops. For each day of measurement, the dependence of surface temperature on incident solar radiation (in terms of the number of pixels) was calculated and processed into graphs in MS Excel with the addition of a trend line and the significance of the dependence based on multiple correlation coefficient $R^2$. Because the

surface temperature is affected not only by sunlight but also by the air temperature during the day, a multiple regression analysis of the dependence of the surface temperature on both the incident solar radiation and the sum of the previously measured air temperatures was performed using STATISTICA 12 software and shown in the form of a surface graph.

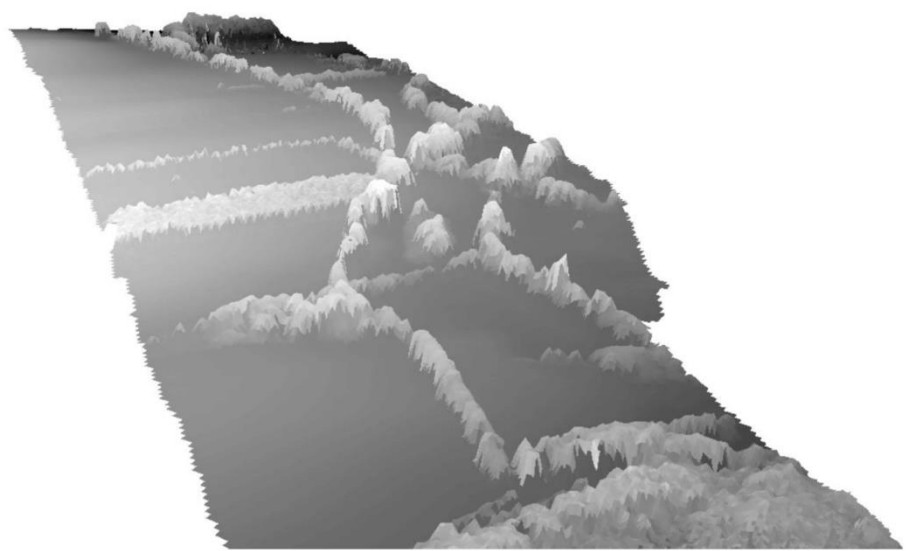

**Figure 3.** Visualization of the DSM.

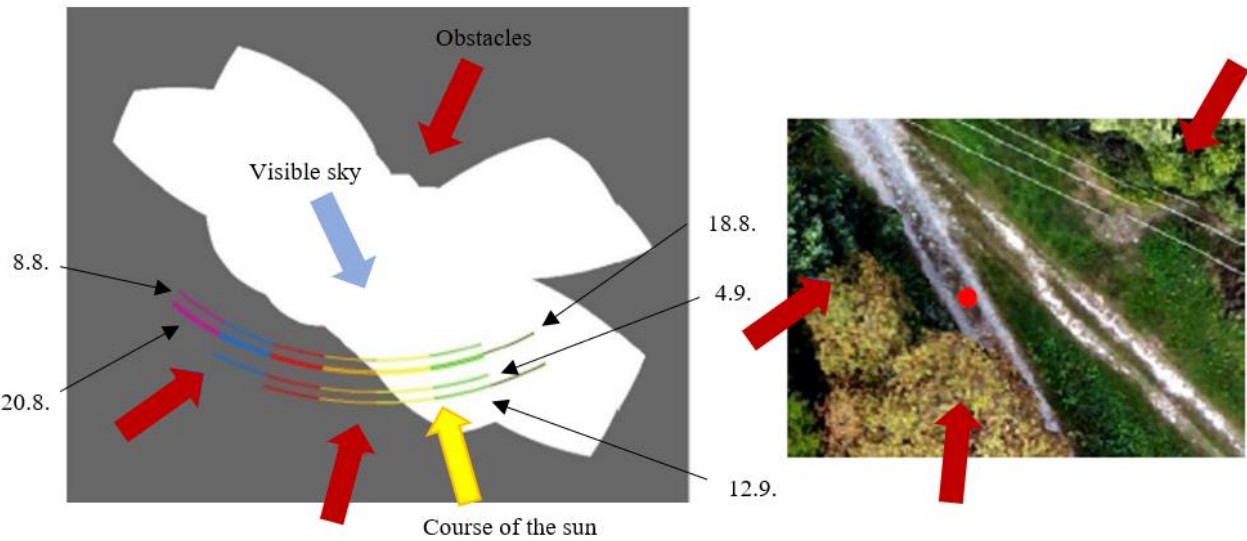

**Figure 4.** Shading of the horizon and the course of the sun. In the figures showing the shading of the horizon above a given point, grey indicates the shading and white means the visible horizon or sunlight. The coloured semicircles mark the course of the sun across the sky during the selected day, which is marked with different colours after each hour. The individual protrusions in grey indicate obstacles (red arrows) that prevent the rays of light from passing to the measured point. These are trees and shrubs that can be recognised when compared to the attached RGB images.

Figure 4 shows also the individual semicircles of the sun's course across the sky on all five days that the site was measured. From above, these were 8 August, 18 August, 20 August, 4 September and 12 September 2019. This scheme is the same for the images of all measured locations. For example, on the third day of measurements, 20 August 2019, measurements began at 11 a.m. (light green), then are shown as yellow at 12 noon, orange at 1 p.m., red at 2 p.m., blue at 3 p.m. and pink at 4 p.m. It can be seen from the figure that,

during the first measurement, there is a point in the sun (white colour) and from the yellow colour—i.e., from 12 noon—the point is in the grey field, so it is shaded.

## 3. Results

Surfaces that are affected by sunlight (marked in light green/grey in Figure 5) have a higher temperature than those in the shade (marked in dark). Temperature also depends on the type of surface and the location on the sunlit asphalt mixture. The sunlit crumbling abrasive layer of the asphalt mixture and the sunlit earth plain have the highest temperatures, up to 37 °C, at an air temperature of around 25 °C (see Figures 6–8). The heat map (Figure 9) was made for an overview of the temperature conditions in the area but was not used for analysis. In contrast, point No. 3 on the asphalt mixture, which was in the shade, reached a temperature of only 20 °C at the same air temperature. The asphalt surface, therefore, has a much higher heat storage capacity than other surfaces and can heat up to temperatures 15 °C higher than the air temperature.

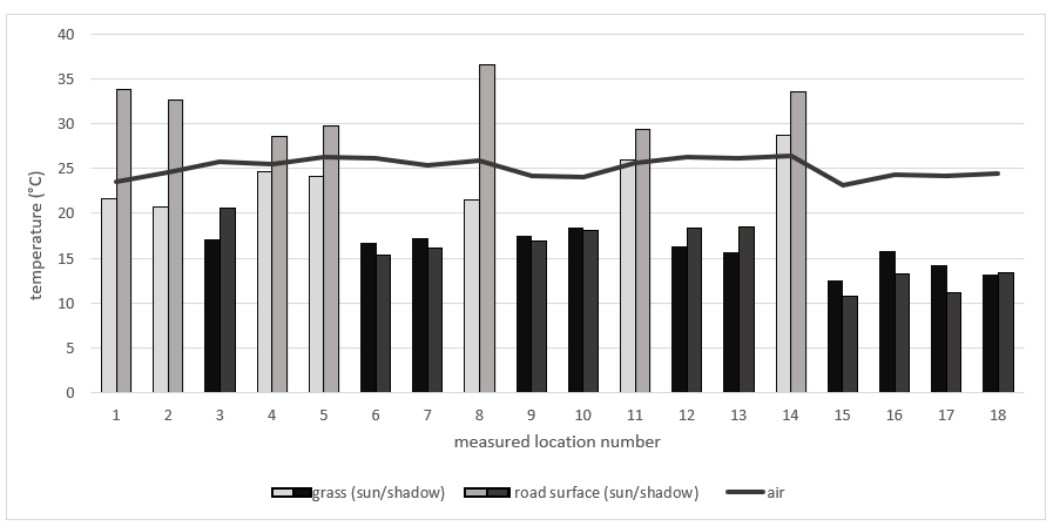

**Figure 5.** Temperature comparison of all measured locations and grass locations with the air temperature at 12 noon on 18 August 2019 (light green/grey—sunlight, dark green/grey—shadow).

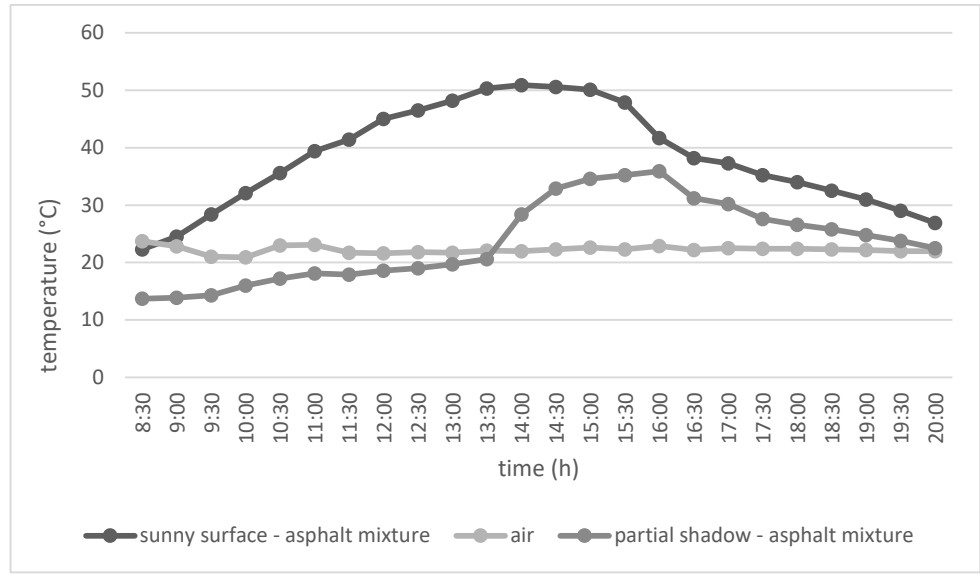

**Figure 6.** Development of temperatures during the day on a sunny/partial shaded surface of asphalt mixture, 5 September 2019.

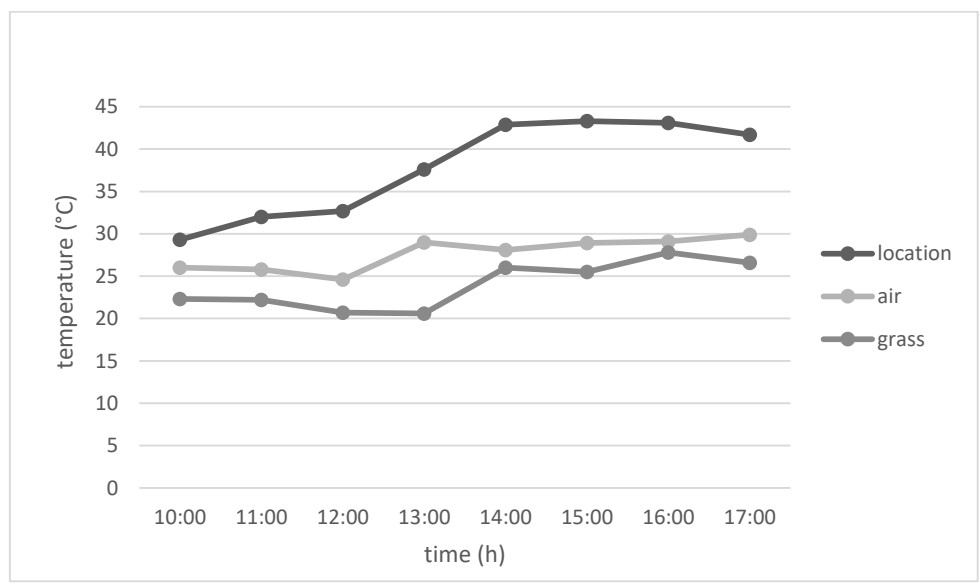

**Figure 7.** Development of temperatures during the day on the surface of asphalt mixture, 18 August 2019.

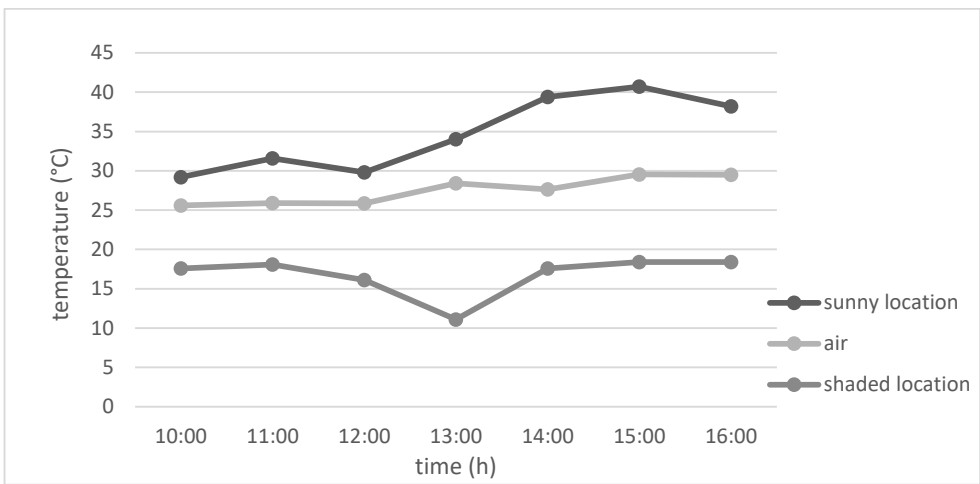

**Figure 8.** Development of temperatures in a sunny/shaded place during the day on the surface of a gravel abrasive layer of crushed aggregate 8/16 mm, which is operationally strengthened, 18 August 2019.

When comparing the temperature of measured locations and grass in Figure 7, it is evident that the grass surface under the same meteorological conditions did not reach such high temperatures, even in direct sunlight as the surface with reinforcement (reinforced surfaces are all measured surfaces except the ground plan). Compared to paved surfaces, there were smaller fluctuations in surface temperature relative to the air temperature on the vegetation cover. The asphalt mixture could accumulate more heat in a shorter time. Compared to the surface with vegetation cover or gravel abrasive layer of crushed aggregate 8/16 mm, which is operationally strengthened (see Figure 8), the asphalt mixture had a faster increase in temperature during the same sun exposure period. It is thus clear that the asphalt mixture has the ability to rapidly accumulate heat, i.e., a sharp rise in temperature, but then its cooling is much slower, in which case twice as long is needed.

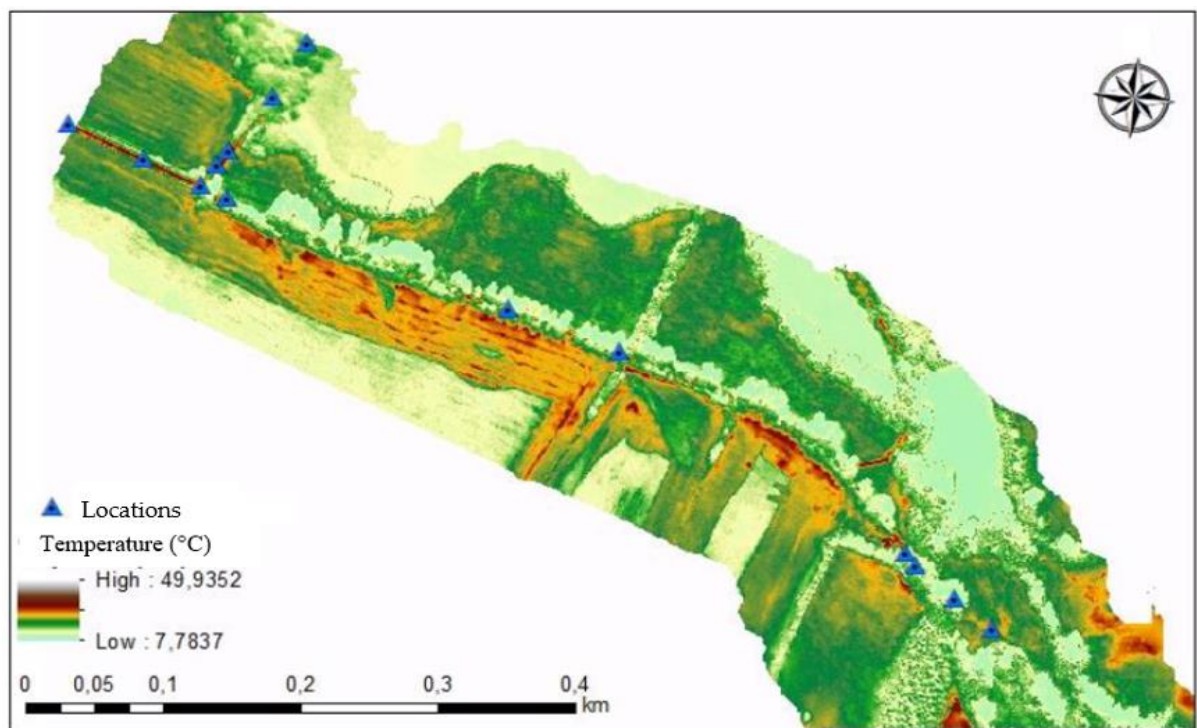

**Figure 9.** Heat map.

According to the measurements, the surface of the asphalt mixture had the greatest ability to accumulate heat and rise in temperature, then the surface with a crumbling abrasive layer of asphalt mixture and the third warmest surface was the earth plane. The coldest measured surface with the lowest heat storage capacity was vibrated gravel and operational reinforcement.

In addition to the dependence of temperature on different types of surface, the dependence of temperature and sun exposure time regardless of the type of surface was solved. This dependence was found only for locations 1–14 because locations 15–18 were not measured during the shooting and thus an accurate digital model of the surface was not created. Figure 10 shows the dependence of the sum of surface temperatures during the measurement period on the percentage of sun exposure during the measurement period on individual days. On most of the monitored days, a significant dependence was found between the sum of surface temperatures during the measurement period and the percentage of sun exposure during this time. On almost all days, an exponential dependence was found between the sum of the surface temperatures during the measurements and the incident sunlight with $R^2$ greater than 0.8. Only on 8 August 2019 was the dependence lower ($R^2 = 0.66$), which could probably be due to clouds during the day. The multiple dependence of the surface temperature on the sum of incident solar radiation and the sum of air temperatures for the previous five hours was also monitored. In multiple regression analysis of all days and all measurement times, the value $R^2 = 0.76$ was reached and thus a significant dependence of the surface temperature on both the incident solar radiation and the air temperature was demonstrated. Estimates of the regression equation parameters are in Table 2. The greater dependence on sun exposure and less on air temperature is also shown in Figure 11.

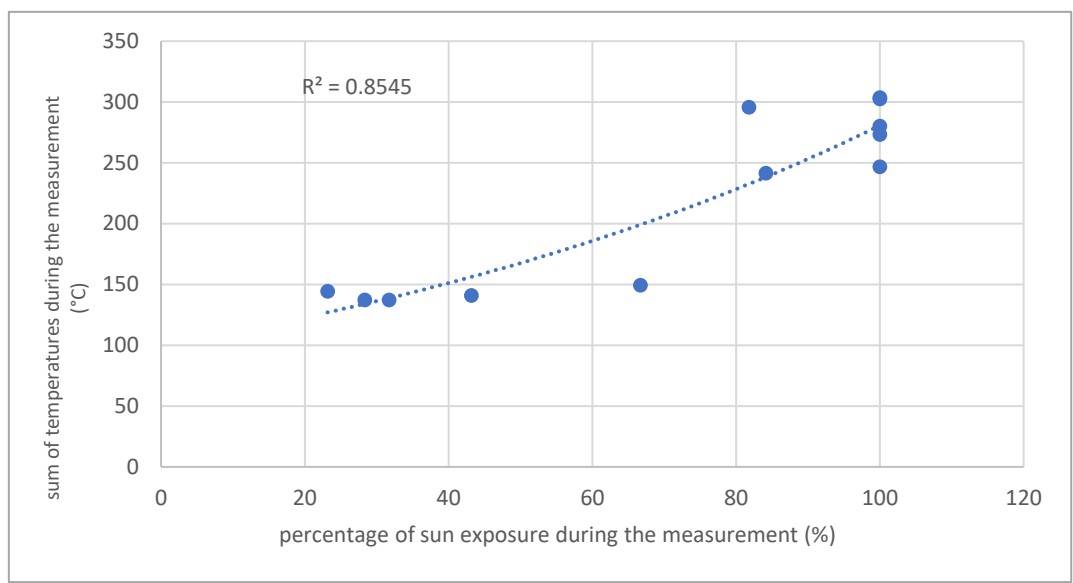

**Figure 10.** Dependence between the sum of temperatures and the percentage of sun exposure during the measurement period on 18 August 2019.

**Table 2.** Multiple regression parameters of model for surface temperature estimation.

| Predictor | Parameter | Standard Deviation | Conclusion | *p*-Value | Lower Bound | Upper Bound |
|---|---|---|---|---|---|---|
| Intercept | −87.05343427 | 26.9783 | Significant | 0.002 | −141.119 | −32.9877 |
| Radiation | 0.01061078 | 0.0011 | Significant | $3.98 \times 10^{-13}$ | 0.0083 | 0.0129 |
| Sumair | 0.952573415 | 0.1627 | Significant | $2.75 \times 10^{-7}$ | 0.6265 | 1.2786 |

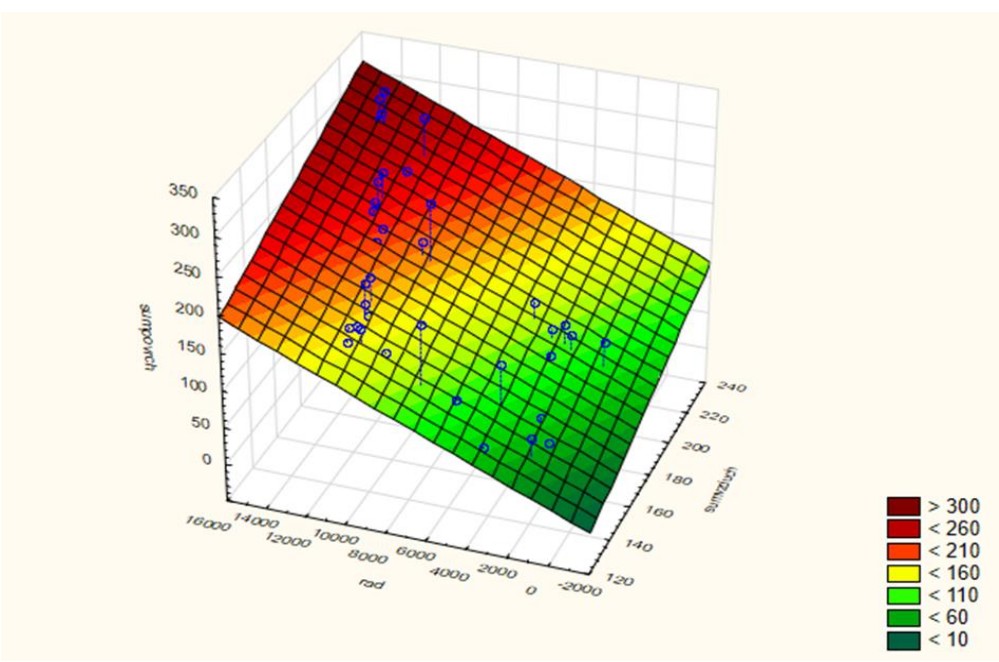

**Figure 11.** Surface graph of linear dependence between the sum of surface temperatures during the monitored days (sumsurface), sun exposure (rad) and the sum of air temperatures (sumair).

The statistical characteristics of the regression model are given in Table 3. Leave-one-out cross-validation was performed using an R script in RStudio software.

**Table 3.** Statistical characteristics of the model for surface temperature estimation.

| Measure | Value |
| --- | --- |
| Multiple correlation coefficient R | 0.87 |
| Coefficient of determination $R^2$ | 0.76 |
| Root mean squared error RMSE | 31.79 |
| Normalised RMSE % | 19.25 |
| Coefficient of determination in cross-validation | 0.74 |
| RMSE in cross-validation | 33.2 |
| NRMSE in cross-validation % | 20.11 |
| Akaike information criterion | 407.25 |
| Maximum variance inflation factor (VIF) | 1.11 |

Normalised mean square error (NRMSE) is RMSE normalized by the mean observed value. During cross-validation, the coefficient of determination and RMSE almost did not differ, and so the regression model can be considered of a high quality. When regression triplet testing was performed (Scott's multicollinearity criterion, Cook-Weisberg heteroskedasticity test, Jarque-Berr normality test, Wald's autocorrelation test, Durbin-Watson autocorrelation test, Sign residue test), only a trend in residues was demonstrated, which could indicate that there was another variable that would explain the data better than the sum of incident sunlight and the sum of air temperatures during the day. The created regression model was created for all types of surface and thus did not include the difference in heat accumulation and loss over time for different types of surface. No other negative factors affecting the credibility of the model were found.

## 4. Discussion

This study had two goals: to demonstrate the influence of different surfaces on their heating and then to demonstrate the influence of vegetation on increasing the thermal well-being of users.

Our measurements showed that the asphalt surface, the surface with a crumbling abrasive layer of asphalt mixture and the natural ground have the greatest ability to accumulate heat and thus increase in temperature. The results of our study are in accordance with published works, e.g., Neog et al. [51], who found lower road surface temperatures in areas with less traffic and more vegetation. Heating ability is affected by the degree of reflectivity, by the albedo. Albedo values of the natural ground are low and range between 0.03 and 0.05 [52], a crumbling abrasive layer of asphalt mixture has an albedo value of around 0.14 [53]. In contrast, the coldest surfaces include vibrated gravel and operational reinforcement. According to [54], who compared the effects of two different solid materials on their surface temperature, it is clear that during the day the surface temperature with a high albedo degree of reflectivity (granite paving) was 4.0 °C lower than the temperature of a material with a low albedo (asphalt). As a result, a material with a high degree of reflectivity can help maintain a lower temperature in its surroundings during the day. During the night, the opposite effect was demonstrated, with granite paving having a higher heat capacity despite its high reflectivity. As a result, it releases more heat after sunset than the asphalt surface, but this effect is a maximum of 0.7 °C. It follows that the solution for day and night reduction of the UHI effect would be to use materials with a high degree of reflectivity and low heat capacity [8].

The phenomenon of warming surfaces can be observed not only in built-up areas but also on hiking and bike trails, where planting and maintaining shading vegetation not only improves the landscape and has many non-productive landscape features [17], but can also increase the attractiveness of trails for tourists. Different types of surface can be

more or less heated and also vegetation along these routes can contribute to their lower temperatures and better thermal comfort for their users [7].

From the results, it can be concluded that for the thermal comfort of road users, bike paths or even city dwellers, surfaces made of vibrated gravel or only with operational reinforcement, which does not heat up so much on warm sunny days and does not affect the temperature of their surroundings, appears to be the most beneficial for a pleasant feeling temperature during hot summer days.

The effect of a heat island can also be regulated by a sufficient number of surfaces with vegetation cover because vegetation absorbs sunlight, using it to drive photosynthesis, which consumes the energy and converts it into assimilates [32], thus maintaining its temperature around 18 °C throughout the day (even at air temperatures around 33 °C) [7]. Due to this low temperature of a surface with vegetation cover, which decreases even lower during the night, greenery can cool the surrounding air [55]. Vegetation in cities and around roads in the landscape not only casts a shadow and thus prevents unnecessary warming of surfaces, but also during the day and then during the night cools its immediate surroundings by photosynthesis and transpiration and thus reduces the air temperature and improves the thermal comfort of people nearby. However, the direct effect of vegetation on the cooling of the surroundings by ecophysiological processes was not the aim of this study, as it would require different methods of measuring and verifying the results.

Due to the shade of nearby plants, the shaded surface does not heat up so much, so it increases the comfort of the users of these roads. Thanks to the shadow, the surfaces experience a slower and lesser temperature rise and thus more easily maintains a constant lower temperature. Using calculations and graphs, an exponential dependence between the sum of surface temperatures during the observed time and the incident solar radiation was found.

During the processing of the results, a significant dependence of the surface temperature on both the incident solar radiation and the air temperature was found. Therefore, it is not possible to assess the surface temperature only by the amount of vegetation present in a given place; other climatic and meteorological conditions at a given time in a given area must be taken into account. Of course, the results also provide other interesting specific information about the types of surface and the effect of solar radiation on their temperature. For example, the temperature of a paved surface in the shade remains almost constant or only rises slightly; but in the sun, we can observe a much faster temperature rise. The measurements also showed that the asphalt mixture could accumulate heat in a short time, while its cooling was slower (it took twice as long as the temperature rise). Climatic and meteorological conditions are constantly changing and thus affect the measured values. Humidity, gusts of wind or a sudden shadow from a passing cloud all have a significant effect on the data collected. Therefore, it cannot be ruled out that with exactly the same measurement procedure, which would take place even under seemingly the same meteorological conditions during the day, different results would be obtained. The type of thermometer used could also cause some inaccuracies in the measurement; more objective results would be provided, for example, by contact temperature sensors permanently located at the measurement locations. On the other hand, from a large body of research and from measurements taken by other authors [7,56], it can be seen that although there were localities (e.g., in the United States) where there are demonstrably different climatic and meteorological conditions, similar results were achieved. The procedure used to evaluate the effect of vegetation on surface temperature and also on the thermal comfort of road users and the subsequent results confirm the need for planting and care of rural roadside greenery.

The results of the study also confirm that modern technologies of data collection using a drone are usable for the analysis of temperature conditions in the countryside, as the created DMPs are usable for modelling the amount of incident solar radiation on the earth's surface. At the same time, it is possible to create thermal maps of the landscape using thermal cameras; however, this method is not optimal for analysing the influence of

vegetation, as it requires repeated campaigns at regular intervals and the heat map is part of the study only to show temperature variability.

Even better results could be achieved using ULS (Unmanned LiDAR Scanning), which could capture the permeability of tree crowns to solar radiation with subsequent voxelization of a point cloud.

## 5. Conclusions

The study confirmed the influence of different types of surface on the temperature of cycle paths, but above all showed the possibility of calculating the dependence of the earth's surface temperature on incident sunlight and the shadow cast by vegetation. It was found and confirmed that shading by woody vegetation had a very fundamental effect on the temperature of any paved surface in its shade and also that a surface covered with vegetation is colder in shaded and sunny conditions than any paved surface. In addition, the evaluation procedure demonstrated the possibility of using drones and DSM to model the course of solar radiation during the day. Created methodology based on the Solar Radiation Graphics tool allowed the performed measurements to be repeated in any area using more accurate methods of surface measurement (e.g., using contact thermometers) with the creation of a longer time series with a higher informative value of the results. The results demonstrate the importance of vegetation in the landscape for reducing surface temperatures and the need for planting and care for greenery in connection with the ongoing global climate change challenge. Therefore, the developed methodology can be used not only for the purpose of evaluating the surface temperatures of cycle paths, but also, for example, for the evaluation of the influence of vegetation in the cultivation of agricultural crops in the form of agroforestry.

**Author Contributions:** Conceptualisation, N.Ž.; methodology, N.Ž.; software, N.Ž.; validation, N.Ž.; formal analysis, N.Ž.; data analysis, T.M., Z.P.; investigation, N.Ž.; resources, N.Ž.; data curation, N.Ž.; writing—original draft preparation, N.Ž.; writing—review and editing, T.M., Z.P.; visualisation, N.Ž.; supervision, T.M.; project administration, Ž.N, Z.P. All authors have read and agreed to the published version of the manuscript.

**Funding:** This research and the APC were funded by the Internal Grant Agency of the Faculty of Forestry and Wood Technology, Mendel University in Brno, Czech Republic, grant number LDF_TP_2019012 "Remote Sensing to Support the Sustainability of Forest Production Under the Condition of Ongoing Climate Change".

**Data Availability Statement:** Not applicable.

**Acknowledgments:** The authors thank the anonymous reviewers for their suggestions and contributions.

**Conflicts of Interest:** The authors declare no conflict of interest.

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
