# Peer review of "The Effects of Roadside Woody Vegetation on the Surface Temperature of Cycle Paths"

_land, doi:10.3390/land10050483_

Round 1

Reviewer 1 Report

Overall the article is well written.
The analyses made are accurate.
I suggest that you compare results with results obtained using satellite images of the study area, to better validate the method.

Author Response

  • I suggest that you compare results with results obtained using satellite images of the study area, to better validate the method.
    • Thank you for your suggestion, Comparison with satellite thermal data is not relevant for this study because of low spatial and temporal resolution. But we have added some description about available satellite data and problems related to this topic.

Reviewer 2 Report

see attachment

Author Response

  • I would recommend expanding the literature to include additional references on surface temperature etc.
    • We tried to expand literature review and include some more references.
  • add location to the list of keywords – Czech Republic
    • it was added
  • I would recommend an English native/editor review and re-read the submission.
    • reviewed by English native editor
  • Abstract - insufficient - the authors should rewrite, or expand the abstract and make it more appealing to the paper. It should be longer, in my opinion, and properly pinpoint all of the paper’s work
    • Abstract rewritten
  • Introduction – insufficient
    • Introduction expanded
  • Results – several shortages
    • All mentioned shortcomings in pictures etc. fixed.
  • Discussion - insufficient.
    • We tried to rewrite discussion to make it more valuable.
  • Conclusion
    • We tried to expand too.

Reviewer 3 Report

Article Name: The effects of accompanying woody vegetation on the surface 2 temperature of cycle paths Abstract: The abstract does not provide any background as to why woody vegetation might influence surface temperature cycles. The knowledge gap is not stated. The methods should be described more thoroughly. A short description of results should be included, and also concluding remark on the method. English editing is advised (for the entire manuscript). Introduction: The introduction presents a case for the differences in thermal cycles for different types of cover within a city or other environments. Even though the discussion is interesting, the section is lengthy. Paragraphs such as the one between lines 98 and 107, might be integrated and reduces in other sections of the intro. For this section, the knowledge gap is not clearly stated. ¿What is the goal? ¿How are the authors going to pursue it? The intent of the study is ambiguous; the authors need to state how are they going to prove their hypothesis, and try a new/old method to test the claim. Materials and Methods: • Figure 1, should be improved and zoomed (unless the area is the rectangle shown in the figure). • L 156 the following is not clear: “The surface temperature, air temperature and vegetation surface temperature were measured 1 m from the edge of the road on permanent grassland”. Is this the sampling method for all locations?? • These section is not clear, regarding the purpose of the study. The authors measured 1) surface temperature on different cover types 2) obtained images from UAV to obtain a snapshot (in time) of surface temperatures for different cover types and 3) measured exposure to sunlight and shading. All the variables might be related, but a clear purpose for each of them must be defined. Results: The results should be presented as summaries on temperatures by cover type. The authors could consider a test observe differences among cover types (instead of figures 7 and 8, the authors might provide a table with deviations). Figure lay out must be corrected (ej. titles within figures might not be necessary, color scheme, etc). Figures 10 to 13, the time of acquisition is different among the graphs, labels are confusing (ej. “point”, “sunny point” and “shaded point”) they should define a cover type according to the authors nomenclature. L253: define “reinforcement”. Figure 14, table 2 and table 3: the figure should not show labels for the “points”, since those labels do not add anything to the analysis. For the tables, results for the test statistic could be summarized in the text.

Author Response

  • The abstract does not provide any background as to why woody vegetation might influence surface temperature cycles. The knowledge gap is not stated. The methods should be described more thoroughly. A short description of results should be included, and also concluding remark on the method.
    • Abstract was rewritten and expanded.
  • English editing is advised (for the entire manuscript).
    • Whole article reviewed by English native editor (The title of the article was a little bit modified by English editor too).
  • Introduction: The introduction presents a case for the differences in thermal cycles for different types of cover within a city or other environments. Even though the discussion is interesting, the section is lengthy. Paragraphs such as the one between lines 98 and 107, might be integrated and reduces in other sections of the intro. For this section, the knowledge gap is not clearly stated. ¿What is the goal? ¿How are the authors going to pursue it? The intent of the study is ambiguous; the authors need to state how are they going to prove their hypothesis, and try a new/old method to test the claim.
    • We tried to shorten some paragraphs in Introduction but other reviewers wanted to expand and add some more information about satellite thermal imaging and UAVs. We add goals of the study and basic principles how can be achieved.
  • Materials and Methods: • Figure 1, should be improved and zoomed (unless the area is the rectangle shown in the figure). • L 156à the following is not clear: “The surface temperature, air temperature and vegetation surface temperature were measured 1 m from the edge of the road on permanent grassland”. Is this the sampling method for all locations?? • These section is not clear, regarding the purpose of the study. The authors measured 1) surface temperature on different cover types 2) obtained images from UAV to obtain a snapshot (in time) of surface temperatures for different cover types and 3) measured exposure to sunlight and shading. All the variables might be related, but a clear purpose for each of them must be defined.
    • We improved and zoomed Figure 1. We tried to explain more correctly the purpose of the study (heat map was not used for analyses)
  • Results: The results should be presented as summaries on temperatures by cover type. The authors could consider a test observe differences among cover types (instead of figures 7 and 8, the authors might provide a table with deviations). Figure lay out must be corrected (ej. titles within figures might not be necessary, color scheme, etc). Figures 10 to 13, the time of acquisition is different among the graphs, labels are confusing (ej. “point”, “sunny point” and “shaded point”) they should define a cover type according to the authors nomenclature. L253: define “reinforcement”. Figure 14, table 2 and table 3: the figure should not show labels for the “points”, since those labels do not add anything to the analysis. For the tables, results for the test statistic could be summarized in the text.
    • We add one figure as a result of ANOVA test of mean temperatures of groups, we redesigned some other figures and their labels too. In figures 10 to 13, the time of acqusition was really different so we couldn´t change. Labels for points removed. Results of statistics summarized in text.

Round 2

Reviewer 3 Report

Figure 13 might not be necessary, since there are only two treatments.

Titles in axis should be revised sometimes there are caps (and sometimes not), fonts seem different, some titles can be shortened.

Colors on figures might not be necessary if there are line trends (there are only three trends in some of them, and different monochromatic line stiles might be enough to depict this trends).

Figure 4, 5 and 6 can be condensed into a single diagram, and use the photo in figure 6 to exemplify.

Author Response

Thank you for your valuable advice and comments. We modified the manuscript, deleted image no. 13, unified the axis labels and merged the mentioned images.